# A Comprehensive Review on Electrocatalytic Applications of 2D Metallenes

**DOI:** 10.3390/nano13222966

**Published:** 2023-11-17

**Authors:** Mohamed A. Basyooni-M. Kabatas

**Affiliations:** 1Department of Precision and Microsystems Engineering, Delft University of Technology, Mekelweg 2, 2628 CD Delft, The Netherlands; m.kabatas@tudelft.nl or m.a.basyooni@gmail.com; 2Department of Nanotechnology and Advanced Materials, Graduate School of Applied and Natural Science, Selçuk University, Konya 42030, Turkey

**Keywords:** 2D metals, metallenes, electrocatalysts, atomically thin structure, electrochemical processes

## Abstract

This review introduces metallenes, a cutting-edge form of atomically thin two-dimensional (2D) metals, gaining attention in energy and catalysis. Their unique physicochemical and electronic properties make them promising for applications like catalysis. Metallenes stand out due to their abundance of under-coordinated metal atoms, enhancing the catalytic potential by improving atomic utilization and intrinsic activity. This review explores the utility of 2D metals as electrocatalysts in sustainable energy conversion, focusing on the Oxygen Evolution Reaction, Oxygen Reduction Reaction, Fuel Oxidation Reaction, and Carbon Dioxide Reduction Reaction. Aimed at researchers in nanomaterials and energy, the review is a comprehensive resource for unlocking the potential of 2D metals in creating a sustainable energy landscape.

## 1. Introduction

Fossil fuel consumption has increased globally due to rising energy demand, which is problematic because of the limited supply of these resources and the serious environmental dangers they entail [1,2]. In recent decades, there has been much promotion of wind, solar, and hydroelectric power as feasible and eco-friendly substitutes for fossil fuels [3]. However, the distribution of these environmentally beneficial resources is limited over time, which is why developing highly efficient energy conversion and storage technologies is essential [3,4]. It has been shown that hydrogen, a low-carbon, high-energy-density fuel, is a feasible medium for storing energy generated from these renewable resources [5,6]. Future fuels that are thought to be promising include hydrogen [7]. Among the many H_2_ generation technologies, electro- and photo-catalytic electro-catalytic water splitting show the most promise due to their affordability and environmental sustainability [8].

Electrocatalysis serves as a fundamental pillar in the field of energy conversion and storage. It plays a pivotal role in promoting chemical reactions that involve electricity, offering significant potential for advancing various technologies. This intricate branch of electrochemistry holds the promise of revolutionizing energy conversion processes, offering a pathway to cleaner, more efficient, and environmentally conscious technologies. Electrocatalysis, at its essence, involves the judicious use of catalysts to expedite electrochemical reactions [9]. Electrocatalysis involves the use of catalysts to facilitate electrochemical reactions at electrodes. These catalysts act as mediators, lowering the activation energy required for specific reactions, thereby enhancing the reaction rates and overall efficiency. By providing alternative reaction pathways, electrocatalysts promote the desired reactions while minimizing unwanted side reactions. One of the most significant applications of electrocatalysis lies in fuel cells. Fuel cells are electrochemical devices that convert chemical energy directly into electrical energy. Electrocatalysts at the anode and cathode facilitate the oxidation of fuel and reduction of oxygen, respectively, enabling efficient electricity generation. Another crucial area where electrocatalysis is employed is water electrolysis, a process that generates hydrogen and oxygen from water. The use of efficient electrocatalysts helps drive the electrolysis reactions with reduced energy input, making hydrogen production more sustainable and feasible for energy storage and fuel applications. Moreover, electrocatalysis has become increasingly important in environmental remediation. By promoting reactions that break down pollutants and harmful compounds, electrocatalysts contribute to the development of more effective and eco-friendly wastewater treatment processes. In recent years, the field of electrocatalysis has witnessed remarkable advancements owing to extensive research and development efforts. Scientists and engineers continue to explore novel materials, design strategies, and reaction mechanisms to enhance catalytic activity and selectivity. These advancements hold great promise for revolutionizing renewable energy technologies and mitigating environmental challenges.

Since the 2004 discovery of monolayer graphene, two-dimensional (2D) materials have garnered increased attention in research as co-catalysts to enhance photo-catalytic H_2_ generation [10,11]. Because of the unique electron confinement present in two dimensions—a feature not possible in other classes of nanomaterials [12,13,14,15,16] or their bulk counterparts—materials with an ultrathin 2D structure exhibit unique and unprecedented chemical, electronic, and catalytic properties [17,18,19]. There has been a significant surge in the 2D material family since the first 2D nanocrystal, graphene, was exfoliated [20]. Graphene analogs, a remarkable class of two-dimensional materials characterized by atomic layers, are fundamentally significant for applications involving surfaces because of their extremely high surface-to-volume ratio and consequently optimized atomic use efficiency. Among the 2D materials with good catalytic, electrical [21,22], and optoelectronic features [23,24,25] are transition metal dichalcogenides (e.g., MoS_2_ [26], WS_2_ [27], g-C_3_N_4_ [28], layered double hydroxides [29]). Moreover, the two-dimensional form reduces the distance that charge carriers must travel to reach the reaction site [30], preventing charge carrier recombination and enhancing photocatalytic efficiency.

With an abundance of active sites and considerable structural flexibility, 2D metallenes combine the benefits of 2D structures with the intrinsic properties of metals. As a result, they frequently exhibit higher electrocatalytic performance compared to their zero-, one-, and three-dimensional equivalents [31,32]. In particular, metallenes have the following benefits: (i) anisotropic nanosheets increase intrinsic catalytic activity and atomic use efficiency by providing a multitude of corners, steps, edges, and flaws; (ii) large solid–liquid or solid–gas interfacial areas combined with short transfer distances allow for fast charge and mass transfer; (iii) nearly all of the catalysts’ surfaces can interact with the electrolyte/reactant and participate in the reactions; and (iv) their well-defined and straightforward structure makes an ideal model for further modifying and elucidating the catalytic mechanism [12]. Due to these outstanding qualities, there is a great deal of interest in researching 2D metallenes as high-efficiency and reasonably priced electrocatalysts. These recent developments motivate us to compile a list of the components and processes needed to create 2D metallenes as well as their benefits for electrocatalysis, particularly in energy conversion processes. We highlight some of the most significant developments in the production, characterization, and electrocatalytic uses of metallenes in this review.

## 2. Three-Dimensional and 2D Metals as Electrocatalysts

Traditional 3D metal catalysts have been widely used in various industries for different purposes, like chemical synthesis and pollution control. However, they come with some challenges that have led researchers to explore other catalytic materials. Some of these challenges include the need for better activity and selectivity in specific reactions, limitations in the surface area leading to slower reaction rates, high cost due to scarce metals and complex production processes, potential instability and leaching of active metal species, susceptibility to poisoning or deactivation, and difficulties in operating under harsh conditions. Additionally, environmental concerns arise due to the toxicity and non-renewable nature of some metal catalysts. To address these issues, scientists are investigating alternative catalytic materials like nanomaterials, metal–organic frameworks (MOFs), 2D metals, and single-atom catalysts, which offer unique properties and potential advantages over traditional 3D metal catalysts [33]. By overcoming these challenges, we can develop more efficient and sustainable catalytic technologies for various applications. Among these materials, 2D metal metallenes look interesting for potential applications.

The main challenge in electrocatalysis lies in optimizing the binding energies of reaction intermediates, following Sabatier’s principle, to achieve better catalytic properties and minimize excessive overpotentials. Noble metals like Pt, Ru, and Ir have been essential reference points due to their near-zero overpotential catalytic activities [34,35]. However, these metals are expensive, prompting research into cheaper alternatives based on transition metals. Although transition metals can be explored as substitutes, they often come with higher overpotentials as a trade-off for activity [36]. A crucial aspect of catalyst design is enhancing the intrinsic activity of active sites, allowing for reduced noble metal usage or increased transition metal loading without compromising performance. In parallel, scientists are exploring low-dimensional nanostructures that contain highly catalytically active units. These nanostructures aim to effectively interact with specific molecular reactants, making this a significant focus in current research.

Furthermore, researchers are focusing on developing low-dimensional nanostructures that contain highly active catalytic units and have a strong affinity for specific molecular reactants [37]. This area of investigation has become a crucial research direction. Among the diverse family of low-dimensional nanostructures, such as 0D, 1D, and 2D materials, particular interest is growing in atomically thin 2D materials. These materials offer unique attributes, including a desirable morphology with a high specific surface area, density, and accessibility of active sites [38]. Additionally, they exhibit excellent conductivity and unsaturated surface coordination, and they maximize atomic utility at high efficiencies. The combination of these exceptional physicochemical properties leads to superior functionality, which significantly impacts the mass transport of reactants toward the highly active centers. As a result, these materials demonstrate outstanding catalytic activity, making them promising candidates for various applications.

In recent years, researchers have been extensively exploring the morphology of metals at the nanoscale, examining their size and structure for their potential applications in various fields like catalysis, sensors, photonics, and biomedicine. In the realm of catalysis, it has been established that customizing the size and shape of metallic nanomaterials can significantly impact their catalytic characteristics, including activity, selectivity, and stability [39]. Increased volume-to-surface-area ratios, improved reactant interaction, the exposure of particular crystal planes, and the existence of more reactive surface sites are all responsible for the gains in catalytic activity. Similar to graphene, atomically thin 2D compounds have recently attracted a lot of interest as potential catalysts. These materials are called metallenes. These materials have a thickness of less than 5 nm and are made completely of metal atoms [40,41]. Metallenes are attractive for catalytic applications because of their fascinating characteristics. As an example, they exhibit exceptional electron transport properties that are on par with or even better than those of other two-dimensional materials such as transition metal dichalcogenides (TMDs) [42,43]. Additionally, the atomically thin layers are rich in under-coordinated metal atoms at the surface and edges, allowing for a maximized specific loading of catalytically active sites [44]. Metallenes are a great option for developing catalytic technologies because of these special qualities.

## 3. Fundamentals of 2D Metals

### 3.1. Understanding 2D Materials and Their Properties

The term “metallene” is akin to “graphene” and signifies a metal sheet with a monolayer thickness. In strict terms, it would mean a metal sheet with only one layer. However, due to challenges in synthesizing freestanding monolayer metal sheets because of thermodynamic instability, obtaining them is exceedingly difficult [32]. Therefore, atomically thin 2D metals with a thickness of less than 5 nm are generally referred to as metallenes. In general, one can modify the compositions, sizes, thicknesses, and surface chemistry of metallenes to modify their distinct properties. Nonetheless, the intrinsic qualities of the metals themselves have the greatest impact on these qualities. This emphasizes the necessity of systematically classifying metallenes into three main groups. Primary group materials such as gallenene (Ga-ene), germanene (Ge-ene), stanene (Sn-ene), plumbene (Pb-ene), antimonene (Sb-ene), and bismuthene (Bi-ene) fall under the group of metallenes. These materials are derived from the main group elements. Transition metal-based metallenes: This group encompasses metallenes based on transition metals. It can be further divided into two subsets: noble metallenes, featuring metals such as gold (Au), palladium (Pd), platinum (Pt), ruthenium (Ru), iridium (Ir), rhodium (Rh), and silver (Ag); and non-noble metallenes, which involve metals like iron (Fe), cobalt (Co), nickel (Ni), copper (Cu), and zinc (Zn). Alloy-based metallenes: the third group involves metallenes formed through alloys, where different types of metals are combined.

Metallenes exhibit unique characteristics that render them very well suited for use in heterogeneous catalysis, in contrast to other non-2D materials and their bulk equivalents. Metallenes have incredibly high surface-to-volume ratios because they are atomically thin. This implies that they can reveal a sizable number of metal atoms that are coordinately unsaturated around their edges and surfaces, which are frequently thought of as active sites [45]. This exposure could potentially enhance their catalytic performance. Edge atoms, in particular, tend to be more coordinatively unsaturated due to defects and dislocations, potentially leading to higher catalytic activity. A crucial advantage of metallenes is that their active sites are fully exposed, allowing for easy functionalization such as ligand modification, alloying, doping, and defect engineering. This adaptability is very helpful in modifying their electronic structures, stability, and physicochemical characteristics, and as a result, their overall catalytic activity. Concurrently, the adsorption of reactants onto the metallene surfaces is greatly enhanced by their large specific surface areas and the metal atoms’ increased surface energy. This interplay of properties makes metallenes particularly promising for catalytic applications [46]. Additionally, the ultra-thin nature of metallenes brings about a substantial reduction in diffusion distances for reactants and products, as well as the distances for charge transfer. This effect accelerates both mass and charge transfer during catalytic processes. Furthermore, some metallenes exhibit an extensive range of light absorption spanning from ultraviolet to near-infrared wavelengths. Applications in photothermal hyperthermia, photocatalysis, photothermal catalysis, and photodynamic treatment are all made possible by this feature. In the context of photocatalysis, it is worth noting that many metallenes cannot directly serve as photocatalysts, due to their lack of bandgap. Nonetheless, there are exceptions like Bi-ene and Sb-ene, as they are semiconductors with bandgaps that vary with thickness [47]. Furthermore, combining these semiconductor photocatalysts with metallenes such as Au and Ag, which possess a pronounced localized surface plasmon resonance (LSPR) effect, has proven to remarkably enhance the transfer and separation of photogenerated carriers. This, in turn, significantly elevates their photocatalytic efficiency [48]. However, a challenge arises due to the high surface energy of metallenes. The surface atoms in these materials are more reactive than those in bulk metals, making them susceptible to oxidation or degradation, especially when exposed to the air. Preserving their metallic nature is thus a prerequisite to retaining their unique properties.

### 3.2. Synthesis Techniques for 2D Metals

As with other 2D materials, there are two primary ways for creating metallene nanosheets: top-down and bottom-up production methods [49,50]. Using mechanical forces and intercalation, top-down procedures include mechanical cleavage, ultrasonic exfoliation, electrochemical exfoliation, and plasma-assisted methods to tear away layers in bulk or powdered samples to produce nanosheets. Conversely, the bottom-up strategy employs chemical techniques like wet chemical synthesis, chemical vapor deposition (CVD), and molecular beam epitaxy. In this case, atoms or molecules are added or assembled in-plane, essentially building up the nanosheets chemically. However, many metallenes display a preference for forming 3D crystals with strong structural symmetry due to their tendency for non-directional metal bonding. This makes their nanostructures containing a high proportion of unsaturated surface atoms unstable. Consequently, directly separating bulk intermetallic layers is challenging through the top-down method. Yet, creating atomically thin metallenes is achievable by introducing surfactants, ligands, or templates. These additives help reduce surface bond energy or adjust anisotropic growth kinetics, resulting in the formation of these thin structures. Alternatively, the bottom-up chemical synthesis method offers an alternative for producing metallenes. This strategy involves carefully controlling the temperature or using the initiator-assisted decomposition of precursors. These decomposed precursors then lead to reduction reactions that form metal nuclei. The nuclei interact with surfactants, ligands, or templates, which inhibit the addition of metal atoms along the longitudinal plane while permitting lateral growth. Through precise control of reaction kinetics, the edges of metallenes can be extended slowly.

#### 3.2.1. Exfoliation

Mechanical cleavage, also referred to as mechanical exfoliation, is a process where a material with layered properties is separated into thinner layers through physical manipulation. This approach is particularly effective for materials like graphene, certain semiconductors with layered structures, and specific metal chalcogenides. The process of mechanical cleavage typically involves placing the bulk material on a substrate or using adhesive tape to hold it in place. The tape is then quickly pulled away, causing the layers of the material to peel off due to the applied mechanical forces. This gradual peeling results in the creation of nanosheets, which can become progressively thinner, even reaching monolayer thickness. One significant advantage of the mechanical cleavage technique is its simplicity. It allows for the direct control of nanosheet thickness by adjusting the number of times the exfoliation process is repeated. Moreover, this method produces nanosheets that are free from defects, which is important for certain applications. However, it is worth noting that the mechanical cleavage method has limitations in terms of scalability and reproducibility. It heavily relies on manual manipulation and might not be suitable for large-scale production, due to its labor-intensive nature.

Recently, successful efforts have been made in the preparation of metallenes based on magnesium and aluminum. Zhang and colleagues introduced a novel technique for exfoliating magnesium into 2D nanocrystals at low temperatures using ultrasound sonication [51]. They incorporated liquid nitrogen treatment to modify the magnesium’s slip system in this process. Similarly, Lu and co-workers employed ultrasound to exfoliate small segments of aluminum foil into 2D aluminum sheets, achieving sizes of around 1 mm in ethylene glycol [52]. Furthermore, using a scalable methodology, Yadav et al. created 2D metal alloys from 3D quasicrystal precursors [53]. They used robust exfoliation in a pressurized ultrasonic reactor to extract 2D structures from the decagonal quasicrystals of bulk A_l66_Co_17_Cu_17_ alloy. The size and thickness of the resulting 2D metallene are important consequences of ultrasonic exfoliation that are mostly dependent on ultrasonic power and duration. There are still difficulties even if ultrasonic exfoliation has the potential to be produced on a large scale and affordably. The current yield of ultrathin 2D nanomaterials, particularly monolayers, remains limited. Moreover, the resulting nanosheets often lack uniformity, and the solvents commonly employed are toxic organic reagents. Addressing these issues has proven to be a challenge yet to be effectively tackled.

Pt nanosheets were created through the exfoliation of layered platinum oxide [54]. Additionally, the process was refined to synthesize monolayer Pt nanosheets, marking the first achievement of such a synthesis by modifying the reduction conditions. These monolayer Pt nanosheets displayed exceptional activity in the Oxygen Reduction Reaction (ORR). Another novel type of catalyst has been introduced, featuring ferromagnetic hexagonal-close-packed (hcp) Co nanosheets (NSs), for the selective electrochemical reduction of CO_2_ to ethanal [55]. Across a range of reduction potentials spanning from −0.2 to −1.0 V (vs. RHE) in a 0.5 M KHCO_3_ solution, these nanosheets consistently yield ethanal as the primary product, with ethanol and methanol formed in smaller amounts. At a reduction potential of −0.4 V, the Faradaic efficiency (FE) for ethanal notably achieves 60%, accompanied by current densities of 5.1 mA cm^−2^ and an MA of 3.4 A g^−1^. The overall FE for the combined products (ethanal/ethanol/methanol) stands at 82%.

A strategy employing liquid exfoliation is proposed to create ultrathin 2D bismuth (Bi) nanosheets, aimed at enhancing the efficiency of electrocatalytic CO_2_ conversion [56] as shown in Figure 1a. The TEM images shown in Figure 1b display the thin nature of the Bi nanosheets, which have a size distribution within the sub-micron range. When examining the HRTEM image in Figure 1b, taken from a specific area marked as point A in Figure 1c, a lattice spacing of 0.395 nm could be measured. This value corresponds to the favored alignment of the Bi (003) plane. Additionally, atomic force microscopy (AFM) has been used to determine that the thickness of the few-layer Bi nanosheets ranged from 1.23 nm to a few nanometers, as shown in Figure 1d. The heightened presence of edge sites on these ultrathin Bi nanosheets compared to bulk Bi played a critical role in enhancing both CO_2_ adsorption and reaction kinetics, thereby significantly facilitating the conversion of CO_2_ to formate (HCOOH/HCOO^−^). Through Density Functional Theory (DFT) calculations, it was determined that the step involving *OCOH formation predominantly occurred on edge sites rather than facet sites, as indicated by the lower Gibbs free energies. Capitalizing on their high conductivity and abundant edge sites, the Bi nanosheets demonstrated a remarkable FE of 86.0% for formate production and an impressive current density of 16.5 mA cm^−2^ at a potential of −1.1 V (vs. RHE), surpassing the performance of bulk Bi. Furthermore, the catalytic activity of the Bi nanosheets remained robust throughout prolonged testing spanning over 10 consecutive hours.

#### 3.2.2. Surfactant-Directed Synthesis

Surfactant-directed synthesis refers to a synthetic approach where surfactant molecules play a crucial role in guiding the formation and growth of materials with specific structures or properties [12]. Surfactants are amphiphilic molecules, meaning they have both hydrophilic (water-attracting) and hydrophobic (water-repelling) parts. In this process, surfactants are strategically introduced into the reaction system. Due to their unique properties, surfactants can interact with different components in the reaction mixture, such as precursors or solvent molecules. The hydrophilic part of the surfactant is attracted to the aqueous phase, while the hydrophobic part tends to avoid water and aligns itself to form micelles or other organized structures. These surfactant assemblies create confined environments or templates that control the nucleation, growth, and arrangement of materials at the nanoscale. By adjusting the type and concentration of surfactants, as well as other reaction conditions, it becomes possible to tailor the size, shape, and morphology of the resulting nanomaterials. Surfactant-directed synthesis offers a versatile and effective way to design and fabricate nanomaterials with desired properties, as the self-assembly of surfactants guides the structure and arrangement of the growing material. This method has been widely used in fields such as nanotechnology, catalysis, and materials science to produce nanoparticles, nanowires, nanotubes, and other nanostructures with controlled characteristics for various applications.

Using less sodium citrate as the reductant, highly pure monodisperse gold nanoplates with regulated sizes were created by reducing hydrogen tetrachloroaurate [57]. When poly(vinyl pyrrolidone) (PVP) was present, the reaction pathway was kinetically regulated. However, the molar ratios of sodium citrate and PVP to hydrogen tetrachloroaurate had a significant impact on the size and geometric shape of the resultant nanoplates. These nanoplates, with dimensions ranging from 80 to 500 nm in width and 10 to 40 nm in thickness, exhibited remarkable single-crystalline properties and displayed efficient surface plasmon absorption within the near-infrared range (700–2000 nm). These gold nanoplates served as essential synthetic nanoblocks, enabling the creation of single-crystalline nanocomponents like nanoscaled gears and intricate nanoscaled letters with precision and versatility.

A simple bottom-up synthetic method with deliberately exposed surface facets is described for creating ultrathin 2D palladium nanosheets (PdNSs) [58]. Their synthetic approach is predicated on the use of amphiphilic functional surfactants’ nanoconfined lamellar mesophases as a template for the formation of PdNSs in an aqueous solution, as shown in Figure 2. They use the long-chain surfactants’ preferential adsorption of functional groups and halide counterions onto certain Pd planes to produce the epitaxial development of the (100)-, (110)-, and (111)-exposed surface facets of ultrathin PdNSs. This general and scalable synthetic approach allows them to precisely control the surface facets of ultrathin 2D PdNSs, offering an avenue to explore facet-dependent catalytic performance. In the context of electrocatalysis for hydrogen evolution reactions (HERs), they demonstrate that (100)-exposed PdNSs exhibit superior catalytic activity and stability compared to (110)- and (111)-exposed versions, as well as bulk Pd counterparts. These findings provide valuable insights that can guide the rational design of surfactant templates for various 2D metal nanosheets.

Another study describes a simple method for synthesizing 2D PdAg alloy nanodendrites that function as high-performance electrocatalysts for ethanol electrooxidation [59]. This process involves co-reducing Pd and Ag precursors in an aqueous solution with the presence of octadecyltrimethylammonium chloride as a structural directing agent. The resulting products exhibit a small thickness (5–7 nm) along with a random in-plane branching structure, leading to increased surface areas and an abundance of undercoordinated sites. They demonstrate improved electrocatalytic activity, as evidenced by a substantial specific current, and they also exhibit exceptional operational stability, as indicated by cycling and chronoamperometric tests conducted during ethanol electrooxidation. Control experiments confirm that these enhancements stem from the combined influence of electronic and structural effects.

Pd stands as the sole metal capable of catalyzing the electrochemical CO_2_ reduction to formate at nearly zero overpotential. Unfortunately, it becomes susceptible to significant poisoning from trace amounts of CO, which is a side product. Furthermore, its stability and selectivity diminish as the overpotential increases. In this study, a solution is presented wherein Pd is alloyed with Cu in the form of 2D nanodendrites to achieve stable and selective formate production [60]. These distinctive bimetallic nanostructures are brought about by carefully controlling a set of experimental parameters that encourage rapid in-plane growth while curbing out-of-plane growth. Through a synergy of electronic and nanostructuring effects, the alloy product demonstrates a remarkable capacity for catalyzing CO_2_ reduction to formate, maintaining notable stability and selectivity even at a cathodic working potential as low as −0.4 V. Computational simulations support these findings, revealing that Cu atoms play a role in the weakening of *CO adsorption and the stabilization of *OCHO adsorption on adjacent Pd atoms.

#### 3.2.3. Vapor Deposition Techniques

Vapor deposition technology has become quite exciting for making 2D metallenes. It involves carefully depositing metal atoms or molecules onto a surface under special vacuum conditions. Scientists can tweak things like temperature, pressure, and how fast they are depositing to control how the metal layers grow and what they end up looking like. For these 2D metallenes, using vapor deposition lets us make super thin layers of metals in a controlled way. There are different methods for vapor deposition, like molecular beam epitaxy (MBE) and CVD, each with its strong points. MBE lets us place metal atoms right onto the surface, giving us pure and well-structured crystals. CVD, on the other hand, works by reacting metal-containing gases on the surface, which helps with covering larger areas. One of the cool things about vapor deposition is that it is good for making a lot of 2D metallene material with the properties we want. It is also reproducible, meaning we can achieve the same results over and over again. This method has already found uses in electronics, making catalysts, and even in optoelectronics because it can help create custom-made 2D metal structures that perform well.

On the substrate, ultrathin 2D nanosheets can be obtained by setting up the right experimental parameters. With the use of UHV-CVD (ultra-high-vacuum chemical vapor deposition) technology, germanium nanosheets were created on Si(100) wafer substrates, and an annealing procedure produced thin-layered germanene with high crystallinity [61]. PVD is the process of physically depositing ions into a collective surface while operating in a vacuum. Millimeter-scale single-phase antimonene nanosheets were obtained on controllable SiO_2_ media substrates through a physical vapor deposition process, as reported by Kuriakose and colleagues [62]. By manipulating temperature gradients in the deposition tube, the production of large-area nanosheets or single-phase antimonene crystals could be tailored as required. Vapor deposition technology carries several advantages over alternative methods for 2D materials synthesis. The deposition rate can be regulated by adjusting process parameters, including gas pressure, gas flow rate, heating rate, and holding time. However, it is important to note that vapor deposition often necessitates elevated temperatures and inert gas conditions, leading to higher production costs.

Wu and their team came up with a method to make antimonene on a PdTe_2_ substrate [63]. They used a super high-tech setup called MBE in an ultrahigh vacuum chamber. First, they split the PdTe_2_ substrate to make single crystals under extremely strong forces. Then, they let the antimony settle on the freshly cut PdTe_2_ and evaporate there. Niu and the gang used the same MBE trick to directly create good 2D antimonene on a special copper oxide surface [64]. They started with a piece of Cu(111) crystal to create a smooth copper oxide layer. Then, they put Sb on this layer at room temperature from a super clean boron nitride crucible. Finally, they performed a high-pressure heat treatment to obtain the thin antimonene layer. Fortin-Deschenes and his crew showed a way to make antimonene on a germanium surface using a process called solid-source MBE [65]. By combining theory and super zoomed-in imaging, they found out that the antimonene they made was good-quality and stable under different conditions. Walker and his team used MBE to grow a thin bismuth film on Si(111), and then they moved it to another silicon piece coated with special epoxy resin using a tricky method called double cantilever fracture technology [66]. The transferred film retained its electrical, optical, and structural qualities just like the original. This MBE technique is expensive and needs special vacuum and pressure conditions. But because it produces a high quality 2D thin films over large scale with high quality, it is favored in some situations.

#### 3.2.4. Advantages and Disadvantages of Preparation Methods of Metallenes

The main advantages and disadvantages of the preparation methods of metallenes are summarized in Table 1.

## 4. Applications of Metallenes in Heterogeneous Catalysis

Metallenes have gained attention as strongly desired heterogeneous catalysts due to their outstanding physicochemical properties, which include a large specific surface area, broad exposure to unsaturated metal sites, and the presence of defective sites. In this context, we aim to shed light on the diverse applications of metallenes across heterogeneous catalysis domains, encompassing areas like heterogeneous organic catalysis, photocatalysis, and electrocatalysis. This exploration is crucial in illustrating the distinct advantages metallenes hold over traditional metal-based catalysts. Notably, considering the promising horizons they offer in terms of energy conversion and storage, we delve into the significant role of metallenes in electrocatalysis within this section.

### 4.1. Oxygen Evolution Reaction

In proton exchange membrane (PEM) electrolyzers, the Oxygen Evolution Reaction (OER) process is commonly regarded as a limiting factor. This is due to its slower reaction rates and higher excess energy requirements when compared to the hydrogen evolution reaction (HER). Despite extensive research into transition metal-based electrocatalysts as cost-effective and efficient alternatives to noble metal-based ones, the most advanced OER electrocatalysts still rely on precious metals like Ir [67] and Ru [68]. Recently, researchers have highlighted the promising performance of 2D configurations of Ir and Ru electrocatalysts, including their alloys, as catalysts for the OER. An illustrative example is the work by Zhao and colleagues [69], who presented structured Ir nanosheets in a 2D arrangement. These nanosheets exhibited remarkable OER capabilities due to their distinct structural advantages, such as well-organized mesostructures, mixed-valence states, and a high presence of electrophilic oxygen species. The optimized Ir−IrO_x_/C catalysts achieved exceptional OER efficiency, with an overpotential of only 198 mV at 10 mA cm^−2^ in an acidic environment. This performance surpassed that of conventional Ir/C catalysts and commercial IrO_2_ materials. Additionally, the researchers developed amorphous Ir nanosheets featuring a significant concentration of unsaturated atoms. These amorphous nanosheets displayed superior OER performance, necessitating only a 255 mV overpotential at 10 mA cm^−2^ in acidic conditions. This achievement was notable when compared to crystalline Ir nanosheets (280 mV at 10 mA cm^−2^) and commercial IrO_2_ catalysts (373 mV at 10 mA cm^−2^) [70]. Using methods such as scanning transmission electron microscopy (STEM) and in situ X-ray absorption fine structure spectroscopy (XAFS), the researchers investigated the structural changes of the amorphous Ir catalyst during the OER. Their research showed that during the OER process, Ir’s valence state remained below +4. The elevated catalytic activity and exceptional stability of the Ir catalyst were facilitated by the preservation of a lower valence state in conjunction with its amorphous structure. Rather than focusing on Ir, researchers have dedicated more attention to the development of efficient electrocatalysts for the OER that are based on ruthenium (Ru). This is driven by the fact that Ru is more affordable compared to other platinum-group metals. However, a significant challenge arises during the OER process due to the generation of soluble RuO_4_, which leads to substantial performance and stability deterioration. In response to this problem, research has shown that adding other metals to the alloy can increase the durability of Ru-based electrocatalysts by strengthening the bonds between Ru and oxygen atoms (Ru−O bonds), which prevents the formation of soluble RuO_4_ species during the OER. An illustrative example is the creation of Mn-doped ultrathin Ru nanosheet branches possessing naturally abundant edges, which exhibited notable electroactivity and endurance in OER scenarios [71]. Both experimental investigations and computational results using DFT have indicated that the presence of manganese (Mn) atoms can modify the electronic structure of Ru sites through the process of doping the Ru crystal lattice. This modification leads to the shortening of Ru-O bonds. Consequently, this alloyed catalyst demonstrated remarkable OER performance, requiring only an ultra-low overpotential of 196 mV at 10 mA cm^−2^ in an acidic environment, and showcased a high level of stability. Furthermore, other 2D bimetallic alloys like channel-rich RuCu nanosheets [72], nanocoral-like RuIr solid solution alloys [73], and IrRh alloy nanosheets [74] have been fabricated. These structures exhibited outstanding adaptability in OER performance when compared to their bulk counterparts.

An electrocatalyst of PdMo bimetallene, consisting of a palladium–molybdenum alloy arranged in a highly curved, sub-nanometer-thick metal nanosheet, has been shown in another work to effectively and sustainably support the ORR and the OER in alkaline electrolytes [39]. Additionally, this bimetallene shows promise as a cathode for lithium–air and zinc–air batteries. PdMo bimetallene has a high atomic utilization and a significant electrochemically active surface area (138.7 square meters per gram of palladium) due to its unique structural characteristics, which are typified by its thin-sheet composition. It therefore shows an MA of 16.37 amperes per milligram of palladium at 0.9 volts in alkaline electrolytes for the ORR in comparison to the reversible hydrogen electrode. This MA is astonishingly higher than those of Pt/C and Pd/C catalysts available on the market by 78 and 327 times, respectively. Remarkably, the catalyst’s performance holds steady even after passing through thirty thousand possible cycles. Using Density Functional Theory calculations, several factors are found to be responsible for PdMo bimetallene’s improved performance.

These include the strain resulting from the curved geometry, the alloying effect, and the quantum size effect because of the thin structure of the nanosheet. Together, these components optimize the system’s ability to bind with oxygen by fine-tuning its electrical structure. Considering the observable characteristics and well-established correlations between the structure and activity of PdMo bimetallene, it is suggested that alternative materials with a metallene structure comparable to PdMo bimetallene may hold great potential for energy electrocatalysis. In another investigation, the emergence of a novel category of 3D Cr-doped Pd metallene nanoribbon (PdCr MNR) assembly is reported [75]. The 3D PdCr MNR structure exhibited a thickness of approximately 4 or 5 atomic layers and was characterized by the presence of Pd sites with high density and low coordination, alongside abundant defects. When evaluated, the 3D PdCr MNR/C composite demonstrated notably improved capabilities in the ORR and the formic acid oxidation reaction (FAOR), surpassing the performance of commercial Pt/C and Pd/C counterparts, and displaying remarkable long-term durability.

Iridium metallene oxide, specifically the 1T phase-IrO_2_, has been reported by Dang et al. [76]. The OER performance of 1T-IrO_2_ is shown in Figure 3. This compound is produced through a synthetic approach that combines mechanochemistry and thermal treatment within a robust alkaline environment. Remarkably, this material exhibits elevated activity in the OER, as evidenced by its low overpotential of 197 millivolts in an acidic electrolyte at 10 milliamperes per geometric square centimeter. Furthermore, the 1T-IrO_2_ displays exceptional durability, demonstrated by negligible degradation even after undergoing 126 h of chronopotentiometry measurement under a high current density of 250 mA cm^−2^ within a proton exchange membrane device. The most active polarization curve is found in 1T-IrO_2_, as demonstrated by Figure 3a’s linear sweep voltammetry (LSV), which requires an ultralow overpotential of 197 mV to achieve a current density of 10 mA cm^−2^. Compared to Rutile-IrO_2_, this overpotential is 100 mV lower, indicating the phase’s significant influence in adjusting the electrochemical performance. In contrast, C-IrO_2_ and C-Ir/C produced greater overpotentials, yielding 276 mV and 311 mV, respectively (Figure 3a). 1T-IrO_2_ exhibited the lowest Tafel slope of 49 mV dec^−1^ when compared to Rutile-IrO2, C-IrO_2_, and C-Ir/C (Figure 3b), indicating a more favorable surface environment for the OER pathway. In comparison to Rutile-IrO2, C-IrO_2,_ and C-Ir/C, 1TeIrO_2_ exhibits the largest geometrical current density at 52.7 mA cm^−2^, which is 9.6, 6.3, and 10.8 times higher (Figure 3c). Rutile-IrO_2_ has a mass activity that is 9.7 times lower than that of 1T-IrO_2_, as Figure 3d illustrates. The computation of the TOF values of several catalysts (Figure 3e,f) is more significant. The TOF values are presented in the following order: 1T-IrO_2_, C-IrO_2_, Rutile-IrO_2_, and C-Ir/C. 1T-IrO_2_ has the highest TOF value. Additionally, 1T-IrO2’s OER performance was assessed using 0.5 M H_2_SO_4_ as the electrolyte. After a 45 h stability test, 1T-IrO_2_ maintained 98% of its original activity under the high current density of 50 mA cm^−2^. This is depicted in Figure 3g. Rutile-IrO_2_, C-IrO_2_, and C-Ir/C, on the other hand, were completely deactivated during the long-term test at the high current density. Figure 3i shows the UPD-based polarization curves and Figure 3h shows the geometric polarization curves of 1T-IrO_2_ before and after the stability test at a high current density of 50 mA cm^−2^. The relatively slight decrease in active-area-based activity in 1T-IrO_2_ indicates that the catalyst drop from the electrode during the stability test is primarily to blame for the OER activity loss.

### 4.2. Oxygen Reduction Reaction

The ORR plays a crucial role in cathodic processes within fuel cells and metal–air batteries. However, its kinetics tend to be sluggish, necessitating efficient ORR catalysts that can minimize overpotentials and enhance energy efficiency. When it comes to designing or predicting effective catalysts, the Sabatier principle comes into play. According to this principle, reactants’ or intermediates’ binding energies on catalysts should be balanced between being too strong and too weak [77]. As a result, a quantitative metric for identifying possible electrocatalysts can be the predicted binding energy derived from Density Functional Theory and established by the electronic structure of the catalysts. It is important to take into account the impact of interfacial electric fields and interfacial solvation effects on electrocatalytic reaction kinetics, as these factors have a significant influence. Consequently, designing catalysts requires the consideration of diverse strategies to regulate the electronic structures of these catalysts. Effects such as ligand modifications, charge transfers, geometric changes, lattice stresses, and their combinations are frequently included in these tactics. When particular examples are examined, the unique structural advantage that metallenes provide in the setting of the ORR becomes apparent. Pd metallenes, for example, have outperformed a commercial benchmark Pt/C catalyst in terms of ORR activity and catalytic stability [78]. This highlights how metallenes can stand out due to their unique characteristics in enhancing catalytic performance.

Because of their suitable oxygen-binding energies, platinum-group metals and their associated alloys have been developed as efficient electrocatalysts for the ORR. These materials exhibit high activity and endurance in acidic environments, with a suitable binding energy between the catalyst surface and O_2_ [79]. Nonetheless, the ORR’s sluggish reaction kinetics at the cathode, stemming from the challenging activation and fragmentation of O=O bonds, result in an excessive reliance on expensive Pt-based catalysts [80]. This dependency hampers the widespread commercial production of fuel cells. To surmount this obstacle, a viable approach involves the construction of 2D noble metal nanosheets. These nanosheets offer an abundance of edges, kinks, and unsaturated atoms that are conducive to the adsorption and activation of O_2_. Consequently, this arrangement facilitates the kinetics of the ORR. For instance, Sugimoto and colleagues have effectively created Pt nanosheets with a thickness of 0.5 nm through the topotactic reduction of platinum oxide nanosheets [81].

Huang and their team undertook the synthesis of porous Pt metallenes by employing the electrochemical erosion of 2D PtTe_2_ nanosheets [82]. These Pt metallenes showed a noticeably higher mass and specific activity in the ORR by a factor of 9.8 and 10.7, respectively, compared to a commercial Pt/C catalyst. This remarkable improvement was ascribed to the electrocatalytic-corrosion-induced introduction of nanopores, boundaries, and vacancies. The number of coordinated Pt sites, which are electron-depleting centers that determine an effective ORR, decreased as a result of these variables. Interestingly, the deformed Pt metallene showed no activity decline and structural changes even after 30,000 cycles. One major obstacle to the commercialization of Pt-based catalysts is their activity degradation, which frequently causes CO poisoning during ORRs. In response, FePt nanosheets with scattered Pt atoms showed better CO tolerance and a seven-fold increase in MA over Pt/C in the ORR [83]. Advantageous electronic conductivity, faster charge transfer, and the most efficient use of atomically distributed Pt atoms over a 2D wrinkled structure are credited with enhanced functioning. The reduced adsorption energy of CO on PtFe, which is made possible by the abundance of nearby OH species generated next to Fe components, explains this exceptional resistance to CO in PtFe. This design helps to remove CO adsorption on Pt.

In a study led by Zhang et al., it is demonstrated that the introduction of carbon doping proves to be effective in augmenting the catalytic activity and durability of metallenes concerning the ORR [84]. By considering PdMo bimetallene as illustrated in Figure 4, following this, both carbon-doped and undoped Pd_95_Mo_5_ bimetallenes were integrated onto carbon black and subjected to evaluation as electrocatalysts for alkaline ORRs. This assessment was conducted in comparison with a commercially available Pt/C catalyst. The cyclic voltammograms (CVs) corresponding to Pd_95_Mo_5_ bimetallene/C and carbon-doped Pd_95_Mo_5_ bimetallene/C were observed to exhibit the distinct characteristic peaks associated with Pd as depicted. It is observed that carbon-doped PdMo bimetallene displays an approximate doubling of both mass and specific activities in ORRs compared to its undoped counterpart. Furthermore, the carbon-doped variant retains approximately three times more molybdenum (Mo) than the non-doped bimetallene after undergoing 30,000 cycles. Through the utilization of X-ray Photoelectron Spectroscopy (XPS) and theoretical simulations, it is unveiled that the incorporation of carbon results in the formation of covalent bonds arising from the hybridization of Mo and C orbitals. This bonding not only serves to stabilize Mo species and counteract Mo oxidation but also optimizes the binding of oxygen. Consequently, this multifaceted enhancement in the catalyst’s stability and performance is achieved.

### 4.3. Fuel Oxidation Reaction

The Fuel Oxidation Reaction (FOR) is the anodic half-reaction that occurs in fuel cells. When combined with electrocatalysts, chemical fuels such as ethylene glycol, methanol, and formic acid serve as promising energy carriers that can generate electricity. This underscores the significance of developing high-efficiency catalysts for FORs, which is pivotal for advancing the industrialization of fuel cells. While a diverse array of materials has been devised as electrocatalysts for FORs, noble metal-based catalysts continue to be of utmost importance due to their exceptional activities [85]. Notably, two specific categories of electrocatalysts are extensively explored for FORs: 2D nanosheets composed of platinum (Pt) and palladium (Pd). These materials have many benefits, such as a large specific surface area, lots of active sites, and quick mass and electron transmission. These characteristics put 2D Pt- and Pd-based nanosheets in the center of the effort to achieve efficient electrocatalysis for FORs. The production of ultrathin triangular Pt nanosheets, consisting of a single layer of ultrasmall nanocrystals (about 3.5 nm in size), was accomplished by Huang’s research group. Remarkably, these nanosheets exhibited superior performance compared to commercial Pt/C catalysts in the context of the methanol oxidation reaction (MOR) [86]. In a recent development, the application of spin engineering to 2D Pd_59_Fe_27_Pt_14_ nanomeshes yielded an electrocatalyst showcasing exceptional activity and stability in the MOR [87]. The authors observed a closely linked relationship between MOR activity and the spin state of Pd_x_Fe_y_Pt_z_. Specifically, as the extent of Fe doping increased within Pd_x_Fe_y_Pt_z_, the magnetic moment associated with each Fe atom displayed a trend resembling a volcano shape. Crucially, it was discovered that this trend matched the comparable trend seen in MOR activity. The use of 2D metallenes as electrocatalysts for the oxidation of C2 fuels, such as ethanol and ethylene glycol, has seen a noticeable increase in interest recently [88]. This attention is spurred by the advantages they offer, including high energy density, low toxicity, high boiling points, and secure storage. To illustrate, a series of bimetallic PdM (M = Zn, Cd, ZnCd) nanosheets with a thickness of less than 5 nm have been developed for the EOR [89]. Particularly noteworthy is the performance of the obtained PdZn metallene, which displayed significantly enhanced electrochemical capabilities. When compared to pure Pd nanosheets and commercial Pd black, the PdZn metallene demonstrated enhancements in MA for the EOR in a 1.0 M NaOH + 1.0 M methanol solution, with improvements of 1.74-fold and 2.97-fold, respectively. The remarkable functionality of 2D PdZn metallene was ascribed to the synergistic impact of ligands and quantum size, which emerged from the ultrathin structure and alloying processes, respectively. Pd-PdSe heterostructured nanosheets with p-d hybridization and a tensile strain effect were also created [90]. This construction exhibited remarkable prowess in ethylene glycol oxidation reaction performance, accompanied by exceptional selectivity in the C1 pathway, within a solution of 1 M KOH + 1 M ethylene glycol. Notably, the specific activity of the Pd-PdSe nanosheets surpassed that of Pd nanosheets and a commercial Pd/C catalyst by factors of 2.5 and 5.5, respectively.

For use in Direct Formic Acid Fuel Cells (DFAFCs), a 2D ultrathin platinum–tellurium alloy metallene, known as PtTe A-ML, is created using a traditional liquid-phase chemical reduction method [91]. The distinctive attributes of high atomic utilization and the alloying effect contribute to PtTe A-ML’s remarkable electrocatalytic performance in the FAOR. More specifically, PtTe A-ML enables the complete realization of the direct oxidation pathway of FAOR as shown in Figure 5. This achievement effectively curtails the generation of undesirable carbon monoxide intermediates while concurrently enhancing the kinetics of the FAOR process. In acidic environments, the FAOR activity of PtTe A-ML surpasses that of commercial Pt and Pd nanocrystals by factors of 43 and 5.6, respectively. In addition, PtTe A-ML has remarkable electrocatalytic ability in the CO oxidation process. The addition of oxygenophilic Te atoms and the ensuing electron transfer between Pt and Te are responsible for this ability. This interaction improves PtTe A-ML’s efficiency in CO oxidation as well as its durability for FAOR.

### 4.4. Carbon Dioxide Reduction Reaction

The unrestrained utilization of fossil fuels and the resulting excess emission of CO_2_ have exacerbated environmental deterioration, presenting a substantial challenge in the form of global warming [92]. The pursuit of a carbon-neutral energy cycle necessitates the development of advanced CO_2_ fixation technologies. However, achieving this goal is arduous, given that CO_2_ is a thermodynamically stable molecule, characterized by the highest oxidation state of carbon [93]. In recent times, significant attention has been directed toward the electrochemical CO_2_ RR as a promising avenue for converting CO_2_ into valuable carbonaceous compounds, thereby addressing the need for CO_2_ fixation. Notably, formate (HCOOH/HCOO^-^) has emerged as a promising candidate, particularly for hydrogen carriers and formic acid (HCOOH) fuel cells [94]. The appeal of formate lies in its attributes of convenient storage and heightened safety. Despite its potential, the widespread application of large-scale CO_2_ RR systems faces challenges stemming from suboptimal catalytic performance, excessive overpotential, and limited energy conversion efficiency [95]. In light of these challenges, there is a pressing need for the systematic design and synthesis of highly efficient electrocatalysts. This imperative arises from the necessity to mitigate the aforementioned issues and advance the practical implementation of CO_2_ RR technology. For the CO_2_ RR, commonly used electrocatalysts are based on various metals such as Zn, Cu, Ag, and Pd. These metals have three subgroups according to the binding affinities of their intermediates and primary products, which makes them electrocatalysts for the CO_2_ RR: (1) Au, Ag, Zn, and Pd are mainly used to produce CO; (2) Sn, Bi, Pb, and In are used to produce formate; and (3) Cu has the capacity to produce multicarbon compounds. The complex multielectron transfer mechanism, the high kinetic barriers of these catalysts, and the fact that HERs exist as byproducts during the CO_2_ RR present difficulties. Together, these elements lead to yield rates and selectivities that are not optimal [12]. When compared to bulk metals, 2D metallic nanosheets have an abundance of unsaturated surface atoms, which promotes CO_2_ diffusion and increases CO_2_ RR’s electrocatalytic efficiency. The production of asymmetrically crumpled Pd nanosheets with exposed (100) facets was a recent development. This was accomplished by converting ultrathin Pd nanosheets that were defect-rich and dominated by (111) facets [96]. This change prompted surface rebuilding, increasing the number of active sites and decreasing CO adsorption strength on the surface in the process. The conversion of CO_2_ to CO was greatly aided by this surface reconstruction, which in turn produced a very high FE of 93% for CO generation on the reconstructed Pd nanosheets at −0.7 V versus RHE. In a similar vein, Ag nanoparticles with an FE of 65.4% displayed a lower CO FE of 96.8% than triangular Ag nanoplates, which were used as catalysts in the CO_2_ RR [97]. This increased edge site presence on Ag nanoplates is responsible for the enhanced catalytic performance in the CO_2_ RR. These edge locations improved the process’s overall efficiency by being favorable for CO_2_ adsorption and the creation of intermediate COOH*.

Rosenthal et al. investigated the use of Bi-based compounds and surface-modified Bi nanoparticles for the CO_2_ RR in acetonitrile with an ionic liquid (IL) serving as the supporting electrolyte. In this context, CO emerged as the principal product [56,98,99]. In scenarios involving a moderate potential, electrodeposited Bi exhibited notable FE in CO production. This proficiency was attributed to the robust stabilization of CO_2_ intermediates facilitated by imidazolium cations. Notably, CO, being a crucial component of syngas, plays a pivotal role as a feedstock for generating olefins through Fischer–Tropsch technology [100]. Nevertheless, the challenge of managing the decomposition of the IL at elevated voltages proved to be complex. Bi-based catalysts’ application potential in the context of the CO_2_ RR was constrained by the decreased stability and increased cost associated with them in organic media when compared to aqueous systems. To produce ultrathin 2D bismuth (Bi) nanosheets that are used to improve the electrocatalytic conversion of CO_2_, a liquid-exfoliation technique is presented [56]. In contrast to bulk Bi, the heightened presence of edge sites on ultrathin Bi nanosheets proves pivotal in the context of CO_2_ adsorption and reaction kinetics. This enhancement notably promotes the conversion of CO_2_ into formate (HCOOH/HCOO^−^). By utilizing DFT calculations, it was established that the formation of *OCOH tends to transpire on edge sites rather than facet sites, a trend corroborated by lower Gibbs free energies. Benefiting from the nanosheets’ high conductivity and plentiful edge sites, an impressive FE of 86.0% was achieved for formate production, alongside a noteworthy current density of 16.5 mA cm^−2^ at −1.1 V (vs. RHE). This performance significantly surpasses that of bulk Bi. Furthermore, the catalytic activity of the Bi nanosheets remained well preserved even after enduring long-term testing over a continuous duration of 10 h. Zhang et al. employed a simplistic one-pot solvothermal method to synthesize copper-doped 2D curved porous Pd metallene nanomaterials [101]. The resulting upgraded catalysts were demonstrated to be robust bifunctional electrocatalysts, simultaneously facilitating a cathodal ORR and anodic formic acid oxidation (FAO). Notably, the developed PdCu metallene demonstrated remarkable attributes such as an excellent half-wave potential (0.943 V vs. RHE) and a high MA of 1.227 A mgPt^−1^ within alkaline solutions. These values surpass those of the commercial Pt/C counterpart by 1.09 and 6.26 times, respectively. Figure 6a displays measurements taken at a scan rate of 50 mV/s in a 0.5 M H_2_SO_4_ + 0.5 M HCOOH electrolyte. The image illustrates that PdCu demonstrates a higher peak current density in comparison to JM 20% Pd/C. Figure 6b demonstrates the negatively shifted onset potential (E_onset_) of PdCu metallene compared to Pd/C, suggesting the superior kinetic efficiency of PdCu metallene and its heightened ability toward FAO. As seen in Figure 6c, PdCu has a much wider MA (905.5 A/mgPd) than commercial Pd/C (328.1 A/mgPd). Chronoamperometry was used to analyze stability, as shown in Figure 6d. The long-term FAO stability of PdCu metallene was evaluated at 0.2 V for 3600 s in a 0.5 M H_2_SO_4_ + 0.5 M HCOOH solution. First, the enhanced concentration of HCOOH molecules is the cause of the higher current density. The build-up of hazardous intermediates such as CO on the catalyst surface is the cause of the following decrease in current density. After a while, these dangerous species’ rates of adsorption and oxidation equalize, which causes the current density to gradually drop. The current density of PdCu metallene is still higher than that of Pd/C after 3600 s. PdCu catalyst’s exceptional FAO catalytic activity makes it a highly promising candidate for use in DFAFCs.

## 5. Electrocatalytic Organic Synthesis

In the context of organic synthesis, electrochemistry has seen an exciting and continuous rebirth in recent years. This increase in interest in the distinctive benefits of electrochemistry has resulted in the development of a growing variety of electrochemical synthesis approaches, including reactivities that are not directly achievable through classic chemical methods alone [102,103]. In particular, the inventive use of radical intermediates to achieve synthetic transformations is particularly suited for single-electron oxidation and reduction processes made possible via electrochemistry. Electrochemistry becomes a potent tactic to address current trends in organic synthesis when the production and downstream reactivity of these intermediates are regulated by catalytic redox mediators. Either an electrode surface (heterogenous electrocatalysis [104]) or molecular mediators as catalytic species (such as transition metal complexes or redox-active organic compounds; commonly referred to as homogenous or molecular electrocatalysis) [105] can be used in electrocatalysis. Systems containing molecular mediators have been referred to as “molecular catalysis for electrochemical reactions”, while the latter scenario, comprising a catalytic electrode, is traditionally defined as “electrocatalysis” [106]. The significance of organic synthesis based on electrocatalysis could be summarized as follows:

### 5.1. C-C and C-X Bond Formation

Electrocatalysis serves as a game-changer in the formation of carbon-carbon (C-C) and carbon-heteroatom (C-X) bonds. The ability to forge intricate molecular structures under mild conditions opens avenues for more sustainable and efficient organic synthesis.

### 5.2. Functional Group Transformations

Electrochemical methods enable selective functional group transformations, providing a level of precision not easily achievable with traditional chemical methods. This selectivity is crucial in the synthesis of complex organic molecules.

### 5.3. Green Chemistry Principles

Electrocatalytic organic synthesis aligns with the principles of green chemistry by minimizing the use of hazardous reagents and reducing waste. The electrochemical processes often require milder conditions, contributing to a more sustainable synthetic approach.

## 6. Outstanding Properties of Electrocatalytic Activity in 2D Metallene

### 6.1. High Surface Area and Active Sites

One key factor contributing to the exceptional electrocatalytic activity of 2D metallene materials is their high surface area. The two-dimensional nature of metallenes allows for a large exposed surface, providing abundant active sites for catalytic reactions. The active sites on the metallene surface play a crucial role in facilitating the adsorption and interaction of reactant molecules, promoting efficient charge transfer during electrocatalysis.

### 6.2. Electronic Structure and Bandgap Engineering

The electronic structure of metallene materials plays a pivotal role in their electrocatalytic performance. The unique arrangement of atoms in the two-dimensional lattice results in distinct electronic properties. Bandgap engineering, wherein the energy difference between the valence and conduction bands is tailored, influences the materials’ ability to facilitate electron transfer during catalytic reactions. This fine-tuning of electronic structure enhances catalytic efficiency.

### 6.3. Catalytic Synergy of Heteroatoms

Metallene materials often incorporate heteroatoms (non-metal atoms) within their structure. The introduction of heteroatoms can create catalytically active sites and modify the electronic structure, influencing the overall catalytic behavior. The synergy between the metal atoms and heteroatoms can lead to enhanced catalytic activity, with the heteroatoms promoting specific reaction pathways or providing additional binding sites for reactants.

### 6.4. Reaction Mechanism and Kinetics

Understanding the detailed reaction mechanism and kinetics on the metallene surface is crucial. The two-dimensional nature allows for a more direct interaction with reactants, affecting the overall reaction pathway. The kinetics of electron transfer and intermediate formation on the metallene surface can be finely tuned, influencing the overall catalytic efficiency. This aspect is particularly important in designing catalysts for specific electrochemical processes.

### 6.5. Stability and Resistance to Poisoning

The stability of metallene materials under electrocatalytic conditions is a critical aspect. Resistance to degradation or poisoning by reaction by-products ensures prolonged and efficient catalytic activity. The inherent stability of metallenes can be attributed to their robust two-dimensional structure, making them resilient to the harsh conditions often encountered during electrocatalysis.

### 6.6. Quantum Effects and Size-Dependent Properties

Quantum effects, arising due to the reduced dimensions of metallene materials, can impart unique properties. Size-dependent effects influence the electronic and catalytic behavior, making smaller metallene structures exhibit distinct characteristics. Exploring the size-dependent properties provides insights into how the quantum nature of metallenes contributes to their outstanding electrocatalytic performance.

## 7. Future Challenges and Opportunities

### 7.1. Toward Enhanced Application of Metallene Electrocatalysts

While the discussion thus far has shed light on the exceptional electrocatalytic activity of 2D metallene materials and their comparison with existing catalysts, it is imperative to acknowledge the challenges that may arise in their practical application and consider future directions for research.

#### 7.1.1. Stability and Durability

One of the critical aspects requiring attention is the stability and durability of metallene materials under real-world operating conditions. Investigating the long-term performance and potential degradation mechanisms will be essential for ensuring the sustained effectiveness of these materials in practical applications.

#### 7.1.2. Scalability and Synthesis

As the field progresses, scalability becomes a pivotal consideration. Examining the scalability of the synthesis methods for metallene materials and exploring cost-effective approaches will be crucial for their widespread adoption in industrial settings.

#### 7.1.3. Multifunctionality

Understanding the potential multifunctionality of metallene materials is another avenue for exploration. Beyond their electrocatalytic prowess, investigating other functionalities or synergistic effects in different applications, such as sensing or energy storage, could open up new possibilities.

### 7.2. Guiding Catalyst Design for Future Research Directions

Moving forward, the focus should extend beyond comparison studies and delve into the tailored design of metallene-based catalysts. This involves the following:

#### 7.2.1. Rational Design Principles

Establishing a set of rational design principles based on the identified structure–activity relationships could guide the development of a new generation of electrocatalysts with enhanced performance and versatility.

#### 7.2.2. Multidisciplinary Approaches

Encouraging collaborations across disciplines, including materials science, physics, and engineering, will facilitate a holistic understanding of metallene electrocatalysts. Such collaborative efforts can lead to innovative solutions and accelerate the translation of fundamental knowledge into practical applications.

#### 7.2.3. Environmental and Economic Considerations

As the field progresses, evaluating the environmental impact and economic feasibility of metallene materials is paramount. Assessing the sustainability of these materials and their potential integration into existing industrial processes will be crucial for the widespread adoption of metallene-based electrocatalysts.

## 8. Conclusions

Recent developments in the use of 2D metallenes in electrochemical energy conversion reactions are compiled in this review. Finally, it should be noted that electrocatalysis plays a critical role in the field of energy conversion and storage. Its applications include fuel cells, water electrolysis, and environmental cleanup. With the progress of research, electrocatalysis is poised to play a pivotal role in molding a sustainable and eco-friendly energy landscape in the coming years. Various synthetic approaches that consider their compositions have been devised to produce 2D metallenes. The outstanding electrocatalytic activity of 2D metallenes is attributed to their unusual properties, which include their unsaturated coordination atoms, enhanced conductivity, and abundance in active sites. Currently, the synthesis of 2D metallenes is limited to laboratory scales due to the costly and time-consuming trial-and-error method involved in their manufacture. One significant obstacle still stands in the way of producing single-layer, high-yield metallenes on an industrial scale. Moreover, there are restrictions on the synthesis of non-noble metal-based metallenes (such as Cu, Co, Ni, Fe, Bi, and Sb), with few successful examples having been documented. This is partly explained by the fact that non-noble metals are more easily oxidized during reactions and that their higher redox potentials call for stricter reaction conditions and more effective reducing agents. Thus, the development of in situ analysis, machine learning, and comprehensive theoretical simulations is essential to revealing the underlying mechanisms controlling their genesis. These developments will make it easier to explore and find 2D metallic materials that are not just noble metals. Unlike their crystalline counterparts, amorphous metal nanosheets have a large number of dangling bonds and a high concentration of unsaturated atoms, which may provide opportunities to increase electrocatalytic performance. Nevertheless, crystalline forms make up the majority of documented 2D metallenes at this time, with amorphous equivalents receiving less attention. Due to the synergistic effects of multiple metal combinations, alloy nanosheets have recently demonstrated improved electrochemical performance when compared to monocomponent-metal-based metallenes. Therefore, it appears that creating high-entropy metallenes—especially ones having an amorphous phase—is a worthwhile endeavor. This trajectory aims to increase the prepared electrocatalysts’ electrocatalytic capabilities while also reducing prices. Furthermore, catalytic capabilities are enhanced through the combination of atomic defect engineering and faceting design. However, the lack of sophisticated synthetic methods demanding careful control over material synthesis makes introducing atomic high-coordinated flaws into ultrathin 2D metallenes a difficult task.

## Figures and Tables

**Figure 1 nanomaterials-13-02966-f001:**
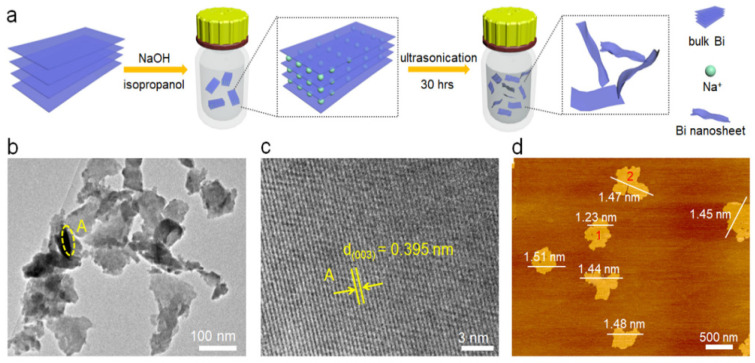
(**a**) Diagram depicting the scalable method for producing ultrathin Bi nanosheets through a liquid-phase exfoliation technique. (**b**) Transmission electron microscope (TEM) depiction and (**c**) associated high-resolution TEM (HRTEM) depiction taken at point A in (**b**). (**d**) Atomic force microscope (AFM) visualization. Adapted with permission from [56]. Copyright (2018) Elsevier.

**Figure 2 nanomaterials-13-02966-f002:**
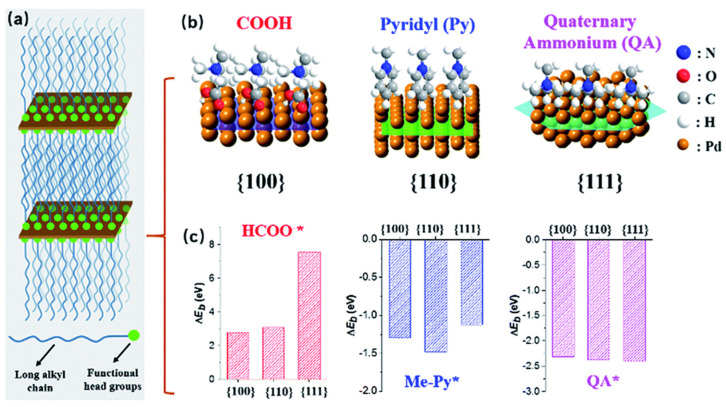
Illustration of the process for synthesizing ultrathin Pd nanosheets with controlled surface facets. (**a**) A summary of how long-chain alkyl surfactants interact to generate layered mesophases, which in turn promote the growth of Pd nanocrystals. (**b**) Illustration of surface interactions and (**c**) corresponding energy differences (ΔE_b_) of functional head groups on different Pd crystal planes, as computed from basic principles [58]. Open access article.

**Figure 3 nanomaterials-13-02966-f003:**
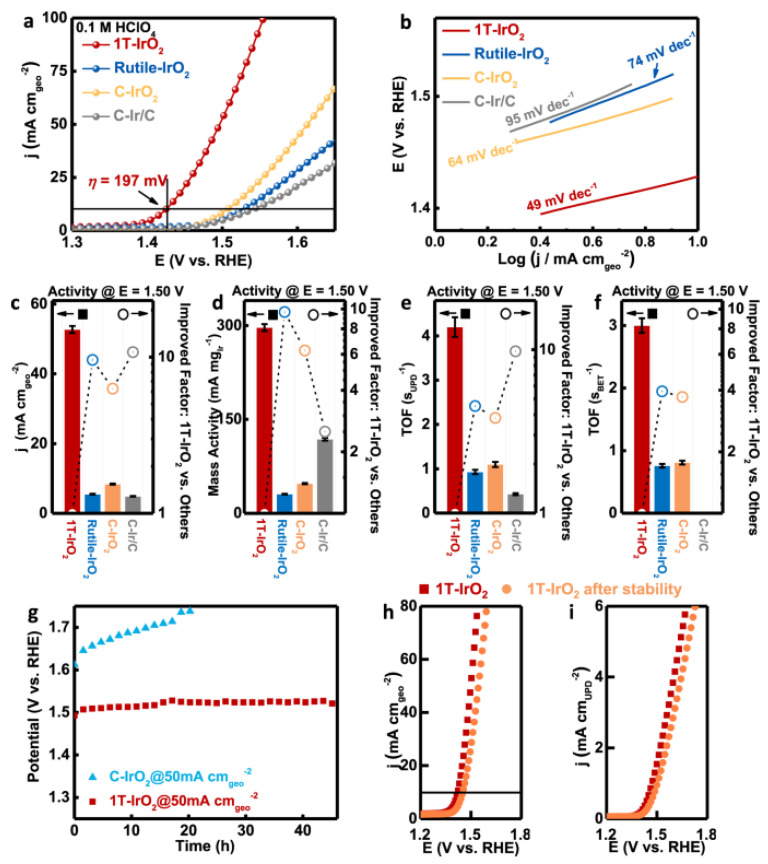
(**a**) In an O_2_-saturated 0.1 mM HClO_4_ electrolyte, polarization curves were produced for 1T-IrO_2_, Rutile-IrO_2_, and commercially available catalysts (C-IrO_2_ and C-Ir/C) after applying iR-correction. 1T-IrO_2_ demonstrated an overpotential of 197 mV in this situation, which allowed for the production of a current density of 10 mA. (**b**) Subsequently, Tafel plots were derived from the polarization curve. (**c**–**f**) The analysis included a detailed comparison between the reversible hydrogen electrodes (RHEs) across 1T-IrO_2_, Rutile-IrO_2_, C-IrO_2_, and C-Ir/C, as well as geometric activities, mass activities, underpotential deposition (UPD)-based turnover frequency (TOF), and Brunauer–Emmett–Teller (BET)-based TOF values at 1.50 V. (**g**) Turning to the performance during chronopotentiometry, the behavior of 1T-IrO_2_ and C-IrO_2_ was assessed under a sustained high constant current density of 50 mA. Impressively, 1T-IrO2 demonstrated its capacity to uphold a notable level of electrochemical activity throughout. (**h**) In addition, the assessment included the geometric polarization curves and (**i**) the UPD-based polarization curves of 1T-IrO_2_ both before and during a stability test that was carried out under challenging circumstances with a high current density of 50 mA. Notably, the alteration in UPD-based active-area-based activity within 1T-IrO_2_ was observed to be minimal. The error bars associated with the data represent the means accompanied by the standard deviation, derived from three replicates (*n* = 3 replicates). Adapted with permission from [76]. Copyright (2021) Nature.

**Figure 4 nanomaterials-13-02966-f004:**
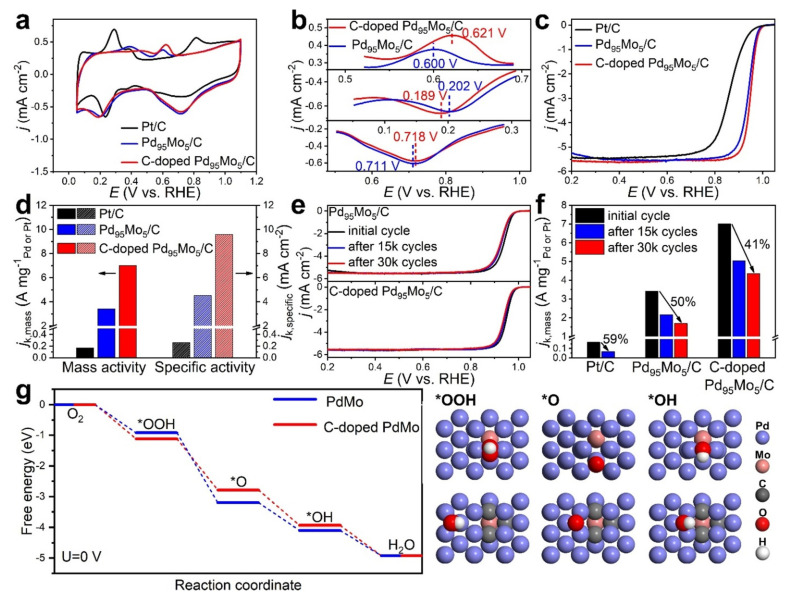
Portrayal of the electrocatalytic performance evaluation of Pd95Mo5 bimetallene/C, carbon-doped Pd95Mo5 bimetallene/C, and commercial Pt/C for the ORR, accompanied by DFT calculations. (**a**) The CVs are presented. (**b**) The CVs are further illustrated with enlarged views of distinct segments. (**c**) ORR polarization curves are depicted. (**d**) The mass activities and specific activities (measured at 0.9 VRHE) of Pd95Mo5 bimetallene/C, carbon-doped Pd95Mo5 bimetallene/C, and Pt/C in a 0.1 M KOH solution are showcased. (**e**) ORR polarization curves are displayed once more. (**f**) The catalysts’ retained mass activities are displayed following exposure to 15,000 or 30,000 potential cycles. (**g**) The relevant adsorption configurations and computed free energy diagrams at 0 V for ORR on PdMo and the carbon-doped PdMo alloy are shown. Adapted with permission from [84]. Copyright 2022 American Chemical Society.

**Figure 5 nanomaterials-13-02966-f005:**
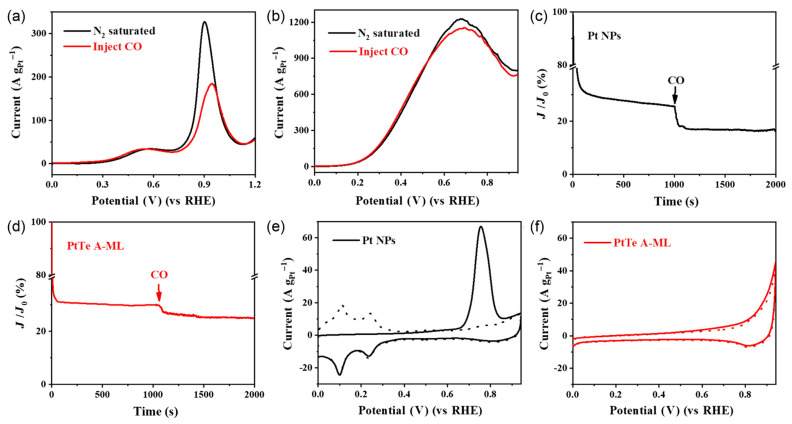
Illustration of the linear sweep voltammetry (LSV) curves of (**a**) commercial Pt nanoparticle and (**b**) PtTe A-ML in electrolytes of N_2_ saturation or CO injection, containing 0.5 M HCOOH and 0.5 M H_2_SO_4_, with a scan rate of 50 mV s^−1^. The response of (**c**) commercial Pt NPs and (**d**) PtTe A-ML to CO injection in 0.5 M H_2_SO_4_ and 0.5 M HCOOH is depicted during chronoamperometry tests. Furthermore, CO-stripping measurements for (**e**) commercial Pt NPs and (**f**) PtTe A-ML are showcased in N_2_-saturated 0.5 M H_2_SO_4_ electrolyte at a scan rate of 50 mV s^−1^ [91]. Open access article.

**Figure 6 nanomaterials-13-02966-f006:**
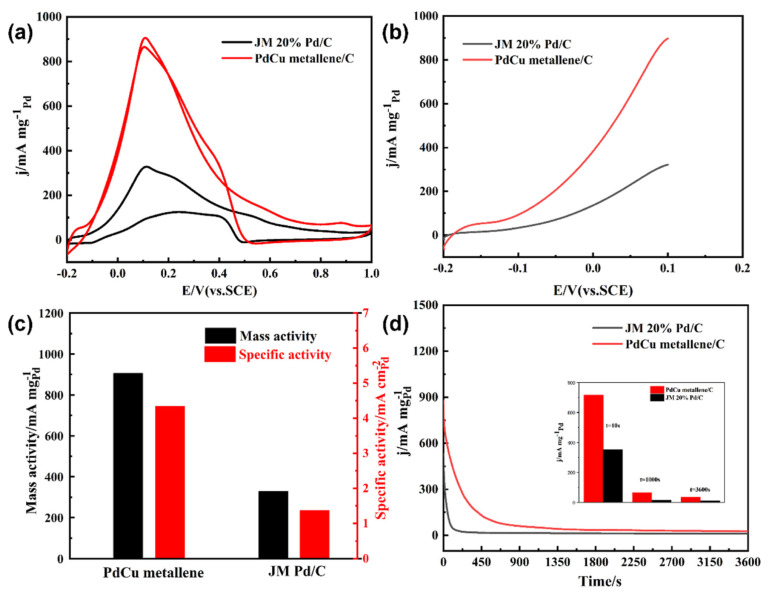
(**a**) The catalysts’ CV curves were recorded in a solution with N_2_ saturation that contained 0.5 M H_2_SO_4_ and 0.5 M HCOOH. (**b**) An enlarged picture of the PdCu metallene and JM 20% Pd/C. The catalysts’ particular activity and (**c**) MA are demonstrated. (**d**) The PdCu metallene and JM 20% Pd/C CA curves are shown at 0.2 V. The catalysts’ current values are shown in the inset of (**d**) after 10, 1000, and 3600 s. Adapted with permission from [101]. Copyright 2023 American Chemical Society.

**Table 1 nanomaterials-13-02966-t001:** Main preparation methods of metallenes.

Method	Advantages	Disadvantages
Top-Down Synthesis	Mechanical cleavage: basic and straightforward technique.	Mechanical cleavage: Challenges in controlling factors like morphology and thickness. Not suitable for large-scale production without further exploration.
	Ultrasonic exfoliation: produces highly pure metallenes with stable morphologies for layered metals.	Ultrasonic exfoliation: struggles to achieve similar effects with non-layered metals.
	Electrochemical exfoliation: suitable for layered metals, providing quick cycles, high yields, and efficiency.	Electrochemical exfoliation: requires further investigation for application to non-layered materials.
	Plasma-assisted processes: controllable morphology and purity.	Plasma-assisted processes: room for improvement in addressing tough reaction conditions and low yields.
Bottom-Up Synthesis	MBE and CVD: high purity, controlled morphology, and good yields.	MBE and CVD: demand harsh reaction conditions, including high temperatures and pressures, contributing to elevated production costs.
	Wet chemical method: uniform, controllable material morphology, high yield, flexibility in reaction conditions and time.	Wet chemical method: Might not guarantee complete purity, leaving behind reaction intermediates. A common challenge in this method.
Summary of Methods	Exfoliation methods: Simplicity, scalability, and relatively low cost. Struggles with precise control over nanosheet properties and yielding high-quality monolayers.	Surfactant-directed synthesis: Tailored structures and improved uniformity through surfactant-guided self-assembly. Challenges with surfactant removal and scalability.
	Vapor deposition techniques: Controlled growth and high-purity products. Specialized equipment, demanding conditions, and elevated costs pose challenges.	
Considerations for Choice	Practical applications: appropriate method selection based on the intended purpose and unique attributes of each technique.	

## Data Availability

Data are contained within the article.

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
