# Peer review of "A Comprehensive Review on Electrocatalytic Applications of 2D Metallenes"

_nanomaterials, 2023, doi:10.3390/nano13222966_

Round 1
Reviewer 1 Report
Comments and Suggestions for Authors
Please see the attached file.

Comments on the Quality of English Language
The quality of English language is adequate.
Author Response
First, we would like to warmly thank the reviewer for accepting our invitation to review and for his valuable contribution to improving the quality of our research under ID: nanomaterials-2716582. We have responded to their comments and concerns below:

Reviewer 2 Report
Comments and Suggestions for Authors
The review is timely and easy to read. I suggest that the author add a section related to electrocatalytic organic synthesis part which are particularly important and were not covered in the review. Moreover, I suggest to extend the introduction with some basics of electrocatalysis using metallenes , in order to make the review more appealing for young researchers.
Author Response

(The authors gave the same response as above.)

Reviewer 3 Report
Comments and Suggestions for Authors
The manuscript, Type of the PaperA Comprehensive Review on Electrocatalytic Applications of 2D Metallenes, by Mohamed A. Basyooni-M. Kabatas discusses interesting work in the direction of Electrocatalytic Applications of 2D Metallenes. This paper can be accepted after minor revisions. The following are few remarks which will be helpful in improving the manuscript further:
1. Abstract: Avoid the use of florid rhetoric, such as engrossing, enthralling, etc., and use only simple statements to present the work done in this review.
2. Introduction: For "1. Introduction to Electrocatalysis and 1.1. Electrocatalysts and Their Role in Energy Conversion". The section has only one paragraph on electrocatalysis, so why use the headings 1 and 1.1? The subsequent part suffers from the same problem in this regard, with unclear headings that are not conducive to readability.
3. The part "2.1. The Promise of 2D Metals as Electrocatalysts" should stand alone, as part 2.1 is not linked to part 2. and there is an error in the logical relationship between the two.
4. For section 3.2.4, the advantages and disadvantages of using different methods for the preparation of metallenes can be presented in the form of a table to make it more straightforward and clear to the readers.
5. For Figure 1., the picture content does not match well with the textual content, which describes DFT calculations on this ultrathin 2D bismuth (Bi) nanosheetst, but the picture does not show the results of the DFT calculations.
6. Some of the content is not presented in sufficient detail. For example, in 4.1 Oxygen Evolution Reaction, there are 4 paragraphs in the section, the first 3 paragraphs introduce the related work in detail, but there is a lack of pictures to show. In paragraph 4, the OER performance of 1T-IrO2 is introduced in a few sentences, but there is no introduction of Figure 3.
Comments on the Quality of English Language
Moderate editing of English language required
Author Response

(The authors gave the same response as above.)
